# Learning from Irregularly Sampled Data with Deep Nadaraya–Watson Kernel Regression Networks (NWNet): Application to Endomicroscopy Image Reconstruction

**Agnieszka Barbara Szczotka**[1], Daniele Ravì[1], Dzhoshkun Ismail Shakir[1],
Stephen P. Pereira[2], Tom Vercauteren[1],
1. Wellcome / EPSRC Centre for Interventional and Surgical Sciences
2. UCL Institute for Liver and Digestive Health
University College London
`agnieszka.szczotka.15@ucl.ac.uk`

## Abstract

Probe-based Confocal Laser Endomicroscopy (pCLE) enables more accurate diagnosis via optical biopsy. pCLE probes relay on of fibres bundles, which generate irregularly sampled signals. Current pCLE reconstruction is based on interpolating irregular signals onto an over-sampled Cartesian grid, using a sub-optimal Delaunay triangulation based linear interpolation scheme. High-quality reconstruction with improved information representation should be possible with the use of Deep Convolutional Neural Networks (CNNs). However, classical CNNs are limited to take as an input only Cartesian images, not irregular data. In this work, we propose to embed Nadaraya-Watson (NW) kernel regression into the CNN framework as a novel trainable CNN layer that allows for processing of irregularly sampled data represented as sparse data on a Cartesian grid. We design a new NWNet architecture in conjunction with examplar-based super-resolution CNN, which allows reconstructing high-quality pCLE images from the irregularly sampled input data. Models were trained on a database of 8806 images from 238 pCLE video sequences. The results were validated through an image quality assessment based on a composition of the following metrics: PSNR, SSIM, GCF. Our analysis indicates that the proposed solution unlocks the potential of CNNs for sparse data processing. NW layer is the main contribution of our end-to-end model performing pCLE image reconstruction directly from sparse imaging input to high-resolution cartesian images. Our method outperforms the reconstruction method in current clinical use.

## 1 Introduction

Probe-based Confocal Laser Endomicroscopy (pCLE) is a recent fibre-based medical imaging modality with utility in a range of clinical indications and organ systems, including gastrointestinal, urological and respiratory tracts [1]. The pCLE probe is an imaging guide used for performing in vivo and in vitro optical biopsy during endoscopic examination.

The pCLE probe relies on a coherent fibre bundle comprised of multiple (>10k) cores that: 1) have variable size and shape; 2) are irregularly distributed across the field of view; 3) display variable light transmission properties, including coupling efficiency and inter-core coupling. The nature of image acquisition through coherent fibre bundles constitutes a source of inherent limitations in pCLE having a direct, negative impact on the image quality including the widely-known honeycomb artefact.

1st Conference on Medical Imaging with Deep Learning (MIDL 2018), Amsterdam, The Netherlands.

The raw data that the pCLE devices produce remains difficult to use for both clinicians and computerised decision support systems as it is modulated by the honeycomb pattern of the fibre bundle and is distorted by a range of artefacts. In the context of pCLE, the irregular sampling domain can be accurately discretised as a set of discrete locations in an oversampled regular grid. Existing pCLE image reconstruction approaches typically use Delaunay triangulation based methods to interpolate irregularly sampled raw signals onto a Cartesian grid, but do not improve quality of images [2]. These methods are themselves prone to generating artefacts, such as triangle edge highlights or additional blur [3, 2]. Incorporating prior information about pCLE images should reduce the uncertainty introduced by the reconstruction process and enable higher quality reconstructions.

To improve the image quality, classically reconstructed pCLE images can be post-processed by restoration and super-resolution techniques. It was shown that state-of-the-art deep learning examplar-based super-resolution (EBSR) techniques improve quality of pCLE images [4]. A limitation in current CNN approaches is that the analysis starts from reconstructed pCLE images over-sampled from the irregularly sampled acquisitions.

Recently, CNNs have shown promising results for medical imaging tasks, but applying these networks to irregularly sampled pCLE data is far from trivial. Existing CNNs rely on a small number of building block types, with shift-invariant convolution being key to unlock the potential of artificial neural networks for image processing and recognition tasks. When applied directly to fibre-bundle-modulated images, whose structure is highly non-shift-invariant, the shift-invariance of CNNs has a detrimental effect. Also, when dealing with small images, such as those produced with fibre bundles, the cropping effect of standard convolutional layers limits the usable depth of the CNNs.

The vast majority of deep machine learning techniques for image data relies on regularly sampled images. Hence there is an unmet need for a unified, computationally-efficient, image reconstruction methodology that compensates for a range of limitations, including irregular sampling, and difficult-to-model artefacts.

We propose a new paradigm by replacing convolution, the cornerstone of state of the art deep learning approaches for classical regularly sampled images, by Nadaraya–Watson kernel regression [5]. We replace the classical CNN layer with a novel trainable Nadaraya–Watson (NW) layer, which can be used as an input layer of any CNN architecture. This NW layer is used to build and validate a unique family of deep learning architectures (NWNets) for irregularly sampled data.

We exploit the irregularly sampled pCLE data for image reconstruction tasks. The main hypothesis of this work is that pCLE image reconstructions can benefit from dedicated deep learning architectures applied directly to the irregularly sampled data. This hypothesis has led us to design an end-to-end pipeline which replaces the classical oversampled reconstruction with NWNets and uses EBSR for regularly sampled data.

The rest of the paper is organised as follows. Section 2 gives a quick overview of the current state of the art methods which enable sparse image data to be used as an input for CNN. Section 3 presents the mathematical and technical details of the NW layer and NWNet networks with the implementation details, used datasets and the training strategy. Section 4 presents quantitative image quality assessment (IQA) for evaluating the performance of NWNets in the context of image reconstruction. Section 5 summarises the contribution of this research to pCLE imaging and deep learning research.

## 2  Related work

**Image reconstruction:** Image reconstruction from sparse signal has been widely studied, and kernel regression techniques have been shown to be a good choice for this task. Takedz et al. notably studied regression methods in the context of denoising and interpolation [6]. Specifically, in the context of pCLE, Vercauteren et al. used kernel regression as a generalisation of Shepard's interpolation with the arbitrary distance kernel [2]. As a part of their mosaicing framework, they implemented reconstruction of scattered pCLE data with Nadaraya–Watson kernel regression using handcrafted Gaussian weighing kernels. They demonstrated that the method efficiently reconstructs pCLE images and reconstructed mosaics at the price of some additional blur.

**Sparse CNN inputs:** Widely used convolution layers have been identified as sub-optimal with regards to sparse data [7]. Several approaches have been proposed to handle sparse data as input to

CNN networks. Much of the available literature on exploring sparsity in the context of CNN input deals with the irregular data in an intuitive way but ad hoc: non-informative pixels are assigned zero, creating an artificial Cartesian image to be input to the network. A similar workaround is to use an additional channel to assign validity to each pixel. For example, Li et al. used that technique and assigned the missing points zero values on a low-resolution image [8], and Kohler et al. passed a binary mask to the network [9]. These solutions suffer from the redundancy in image representation which lowers computational performance and requires the network to learn sparse representation.

Numerous studies have attempted to generalise neural networks to work on arbitrarily structured graphs. For example, Defferrard et al. proposed a graph-based CNN which learns local, stationary, and compositional features for the image classification task [10]. Graph-based CNNs accept a graph as input, and produce a graph as output (graph-in-graph-out). This approach cannot be directly translated to image reconstruction, as this task would involve a graph as input and a reconstructed image as output (graph-in-image-out).

In a recent study, Uhrig et al. proposed a sparse convolutional layer which jointly processes sparse images and sparse masks [7]. The layer is designed to account for missing data during the convolution operation by propagating the sparsity information (encoded in the mask) through the entire network. Their research work on the sparse CNN layer is the most closely related to our work on the NW layer. The critical difference between their approach and ours is that the NW layer performs a methodologically founded Nadaraya–Watson kernel regression, while the sparse CNN layer does not perform a typical regression, but rather uses a non-shared constant kernel for the convolution of the mask.

## 3 Materials and methods

Irregularly sampled data can be represented with an arbitrary approximation quality as sparse data on a fine cartesian grid. Specifically, in the context of pCLE reconstruction, the missing information is interpolated, meaning the reconstructed images are over-sampled, and only a subset of the pixels carries information [4]. As is illustrated in Figure 1, we typically represent the corresponding sparse images by assigning a value of 0 to all non-informative pixels.

Since the position $u, v$ of informative pixels within the sparse image $S$ is given, we can represent the image sparsity in the same space. Let $M$ denote a binary sparsity mask (of the same size as image S) that encodes information such that it takes the values 1 and 0 for the informative and non-informative pixels respectively. A sample sparsity mask with the corresponding sparse pCLE image is also depicted in Figure 1.

### 3.1 NW layer

Convolutional layer is denoted by $f$ as:

$$f_{u,v}(X) = \sum_{i,j=-k}^{k} x_{u+i,v+j} w_{i,j} + b \,, \tag{1}$$

and it takes image $X$ as an input. The image $X$ is represented on a Cartesian grid with pixel coordinates given by $u, v$. The convolution operator considers all image pixels as equally important regardless of position $u, v$. The output of the convolutional layer $f(X)$ is generated by convolving the input image $X$ using weights $w$ and adding a bias $b$. Weights $w$ are defined by a kernel $W$ of size $2k + 1$ along each image dimension.

Since the irregularly distributed pCLE signal is represented on a Cartesian grid as a sparse image, it can be input into any CNN. Nonetheless, if the sparse image $S$ is input to the CNN, the network has to learn not only the function $f(X)$ for informative pixels but also their sparsity, which makes the optimisation of $f(X)$ a difficult task. Here the open challenge is to adapt the CNN to work around the input image sparsity by predicting the missing information. To tackle this problem, we introduce the Nadaraya-Watson kernel regression layer (henceforth referred to as "NW layer").

Sparse pixels of $S$ are related, and this relation can be model. It leads to Nadaraya–Watson kernel regression which models relation of the data points by use of custom kernels to perform local

interpolation. This regression technique can be efficiently implemented using two convolutions and a pixel-wise division, and it was successfully used with a single hand-crafted kernel to reconstruct pCLE images and mosaics [2].

To incorporate Nadaraya–Watson kernel regression into the CNN framework, we define the NW layer as :

$$R_{u,v}(S, M) = \frac{\sum_{i,j=-k}^{k} S_{u+i,v+j} w_{i,j}}{\sum_{i,j=-k}^{k} M_{u+i,v+j} |w_{i,j}|} \tag{2}$$

The NW layer takes as input a sparse image $S$ and a corresponding sparsity mask $M$ that are both convolved with shared kernels $W$. The output of the NW layer are reconstructed feature maps $R$ estimated using a Nadaraya–Watson regression, which we generalise to allow for negative kernel values as described below. The graphical interpretation of NW layer is presented in Figure 1. The mask $M$ can be seen as a probabilistic sparsity map. The input $M \in \{0, 1\}$ is initially a binary mask but arbitrary probabilistic sparsity patterns are then propagated through the NW layers. By convolution with kernels $W$, $M$ is transformed to an approximation of continuous distribution that represents the probability of obtaining $R(u, v)$ given $S(u, v)$.

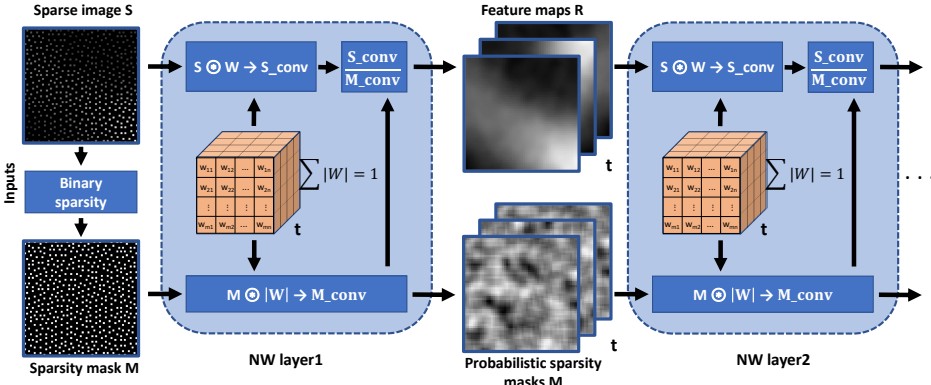

Figure 1: NW layer: the network inputs are the sparse image $S$ and sparsity mask $M$; an NW layer performs two convolutions and a division; the NW layer outputs are reconstructed feature maps $R$ and a probabilistic sparsity mask $M$ after convolutions with kernels $W$.

Classical Nadaraya–Watson kernel regression uses positive kernels, but for flexibility, our NW layer allows for negative values and this generalises the kernel regression. It is necessary for the convolution of the mask $M$ to rely on the absolute value of $W$, as this operation is meant to capture the influence of neighbouring pixels on the predicted values of $R(u, v)$ (see denominator in equation 2). Our proposed NW layer implementation is given as pseudo-code in Algorithm 1. The implementation of the NW layer requires numerical safety measures such as checking against not-a-number (NaN) and infinity values which may arise during the training procedure, and divisions by 0 or very small values from sparsity mask. We also normalise the kernels to $\sum_{i,j=-k}^{k} |w_{i,j}| = 1$ for numerical stability.

## 3.2 NWNet

We conceptualise the NWNet framework as a stack of n NW layers as illustrated in Figure 2. Every NW layer has $t$ unique kernels $W$. The first ($n = 1$) NWNet layer takes as input a sparse image $S$ and a binary mask $M$. Every consecutive NW layer then returns $t$ feature maps $R$ and $t$ sparsity masks $M$ which become the input for the next NW layer. The last NW layer of the NWNet framework returns only $t$ feature maps $R$, and masks $M$ are discarded.

The multiple layers of Nadaraya–Watson kernel regression with learned kernels can be used to generalise standard CNNs for irregularly sampled pCLE data. Given that the deeper classical CNN architecture has better performance, intuitively the same rule should apply to NWNet architectures: the deeper the NWNet, the better the generalisation. We assume that NWNet learns the sparsity of the input data, such that after a few NW layers the output features map can be directed to classical

**Algorithm 1** Implementation of the NW layer

---

1: **function** NW($S$, $M$)
2:     $threshold = 0.0001$
3:     $W = \text{initialisation}(weights\_shape)$
4:     $S_{conv} = S * W$
5:     $M_{conv} = M * |W|$
6:     $W = \frac{|W|}{\sum |W|}$
7:     **if** $M_{conv}(u, v) < threshold$ **then**
8:         $norm(u, v) = \frac{1}{M_{conv}}$
9:     **else**
10:        $norm(u, v) = 0$
11:     **end if**
12:     $R = S_{conv} \cdot norm$
13:     **if** $R(u, v)$ $is$ $NaN$ $or$ $Inf$ **then**
14:        $R(u, v) = 0$
15:     **end if**
16:     **return** $R$, $M$
17: **end function**

---

CNN layers. This in turn facilitates the implementation of end-to-end pipelines that can incorporate sparse inputs by combining NWNet with any classical CNN architecture.

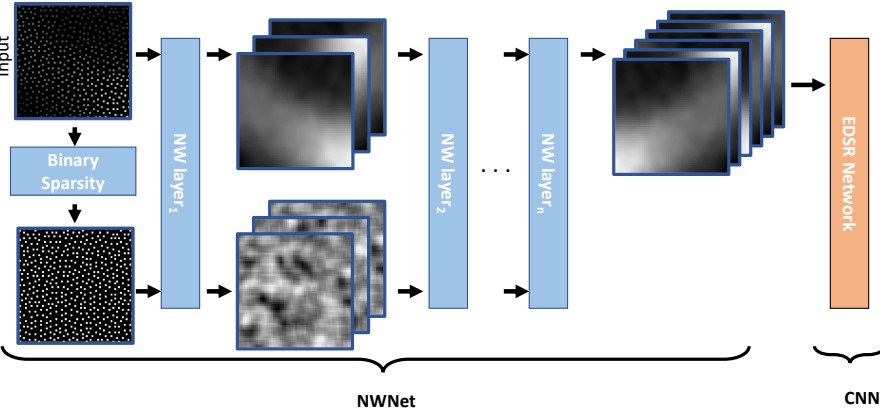

Figure 2: NWNet framework: the NWNet input are the sparse image $S$ and binary sparsity mask $M$; a NWNet is a stack of $n$ NW layers which utilise $M$ to learn the sparsity of $S$; the NWNet output are reconstructed feature maps $R$.

### 3.3 Application to endomicroscopy image reconstruction

NWNet unlocks access to complex deep learning models which can contribute to the less uncertain reconstructions by providing prior information, such such as super-resolving filters trained by exemplar-based super-resolution methods. It was shown in [4] that Enhanced Deep Residual Networks for Single Image Super-Resolution (EDSR) [11] allows for improving the quality of over-sampled Cartesian pCLE images. We decided to build a pCLE reconstructions pipeline by merging NWNet with EDSR. These way, we utilise advantages of both: NW layers enabling existing deep-learning models to perform training on sparse pCLE images, and EDSR as a post-processing method to improve the pCLE reconstruction.

### 3.4 Experiments

**Networks:** We propose five NWNet architectures combined with EDSR as presented in Figure 2. The proposed NWNet differs by the number of kernels and layers as given NWNet(layers, (kernels)).

In the original work EDSR has 16 residual layers. To maintain depth of the entire pipeline, the number of residual layers in EDSR is reduced by the number of NW layers, and donated EDSR(17 - NW layers). The networks are designed as follows:

- three shallow (in terms of NW layers only) architectures with a single NW layer with 32, 64, 128 kernels each:
  i. NWNet(1, (32)) + EDSR(16); ii. NWNet(1, (64)) + EDSR(16); iii. NWNet(1, (128)) + EDSR(16).
- two deep architectures: the first architecture has 2 NW layers with 128 and 64 kernels for each layer respectively: NWNet(2, (128, 64)) + EDSR(15);
  and the second architecture has 3 NW layers with 128, 64, 32 kernels for each layer respectively: NWNet(3, (128, 64, 32)) + EDSR(14).

As a baseline method we used the EDSR network, which takes as input reconstructed pCLE images as presented in [4].

**Dataset**: Networks are trained on synthetic pCLE dataset published in [4]. Three subsets are created as a fraction of the entire dataset: train (70%), validation (15%) and test (15%) sets.

The high-resolution (HR) images are transformed to low-resolution sparse pCLE images (LR) by assigning zeros to all pixels which do not correspond to any fibre signal. These sparse pixels are normalised: $LR = LR - mean_{LR}/std_{LR}$ and $HR = HR - mean_{LR}/std_{LR}$, and scaled in the range [0-1] for every sparse frame individually, without considering non-informative zero pixels. Lastly, for the train and validation sets, non-overlapping $64 \times 64$ sparse patches are extracted from the processed images.

NWNet described in Section 3.2 takes as input a sparse image with the corresponding binary mask. The mask is generated by assigning 1s to where fibre signal is, and 0s to rest of the image space (for the example patch see Figure 1).

The test set, which is not available during training, is built with pCLE full-size original sparse images. It is important to note that ground truth images are not available in the context of pCLE. The synthetic pCLE dataset published in [4] was generated with a registration-based simulation which produces synthetic high-resolution (HR) estimates of ground truth images. The final performance of the NWNets is tested by comparing reconstructed super-resolved (SR) pCLE images with the synthetic HR images.

**Training strategy**: Networks are trained with batch-based stochastic strategy, with a batch size of 32 patches. Our choice for optimiser is RMSprop [12] with a momentum and a learning rate of 0.1 and 0.0005 respectively. Gradient clipping technique is applied to reduce the problem of vanishing gradients with the gradient clipped from -1 to 1.

The kernel size for NWNet layers is 9 (k=3) across each image dimension. The size was chosen based on known distribution of fibres across a Cartesian image to ensure that each convolution would capture more than 10 informative pixels. The weights were initialised with a truncated normal distribution with mean, and standard deviation equal to 0.2 and 0.05 respectively. Additionally, all the weights in NW layer are regularised with L1 and L2 norm, with a scale 0.1 for both.

The models are first trained with the L1 loss. These pre-trained models are further trained with SSIM+L1 loss [13]. This training strategy was chosen based on results presented in [4], which indicates that the best performing models were trained with SSIM+L1. L1 loss is computationally less expensive than SSIM+L1, so networks were pre-trained to shorten training time. The threshold in Algorithm 1 is set empirically to 0.0001.

## 4 Results

The purpose of the experiments proposed in 3.4 was to validate the performance of the super-resolving pipelines (NWNet + EDSR) by assessing the quality of the reconstructed SR images. To measure image quality of the SR pCLE reconstructions we design an image quality assessment (IQA) procedure by combining three complementary metrics. Typically, metrics used to assess the quality of the images are the peak signal-to-noise ratio (PSNR) and the Structural SIMilarity index SSIM [14]. Good image contrast is also desirable for pCLE images, and this was judged based on

Global Contrast Factor (GCF) [15]. GCF is reference-free technique, thus to measure improvement in contrast we computed the differences $\Delta$ of GCF(image): $\Delta\mathrm{GCF}_{\mathrm{LR}} = \mathrm{GCF(SR)} - \mathrm{GCF(LR)}$ and $\Delta\mathrm{GCF}_{\mathrm{HR}} = \mathrm{GCF(SR)} - \mathrm{GCF(HR)}$. If contrast of SR images is improved, $\Delta\mathrm{GCF}$ is a positive.

All mentioned metrics measure different properties of the image. Therefore the total combined score $Tot_{cs}$ was introduced similarly to [4]. $Tot_{cs}$ is defined as arithmetic average score of all the normalised (in range $[0-1]$) metrics: PSNR, SSIM, GCF.

The results computed for the test images are shown in Table 1. It is apparent from this table that NWNet + EDSR give better reconstruction in comparison the base line method. Closer inspection of the table shows that:

1. NWNet + EDSR applied to sparse images outperforms EDSR applied on Cartesian images for contrast measurement, but gives slightly lower results for SSIM and PSNR. The difference of total scores given by $Tot_{cs}$ between EDSR(16) and the simplest NWNet(1, (32)) + EDSR(16) is -1, and the deepest NWNet(3, (128, 64, 32)) + EDSR(14) is -3. When $Tot_{cs}$ is considered every NWNet + EDSR performs better than just EDSR.

2. The $Tot_{cs}$ for NWNet(1, (32)), NWNet(1, (64)), NWNet(1, (128)) increases with the number of kernels, thus it highlights that NWNet-based reconstruction benefits from a larger number of trained kernels. The more surprising observation is that individual changes in the metrics' score are not correlated with the number of kernels.

3. With successive increases in the number of layers (deeper NWNets) we obtain the better quality reconstructed images. $Tot_{cs}$ drops by -4 for EDSR(16) and both deep architectures. The performance of NWNet with 3 layers is slightly better than NWNet with 2 layers, when scores are considered individually.

Overall, these results reveal that the best performing model is NWNet(3, (128, 64, 32)) + EDSR(14), which is the deepest (in terms on NW layers) NWNet out of all proposed models. The results confirm that NW layer is a good choice for image reconstruction and yields increasingly good results on sparse pCLE data.

We provide example reconstructions in figure 3.

Table 1: IQA for NWnets.

| network | SSIM | PSNR | $\Delta\mathrm{GCF}_{\mathrm{LR}}$ | $\Delta\mathrm{GCF}_{\mathrm{HR}}$ | $Tot_{cs}$ |
|---|---|---|---|---|---|
| EDSR(16) | 0.864±0.049 | 25.6±3.0 | 0.97±0.79 | 1.20±0.90 | 0.42±0.20 |
| NW(1, (32)) + EDSR(16) | 0.814±0.070 | 24.5±3.0 | 1.72±0.90 | 1.94±1.10 | 0.43±0.20 |
| NW(1, (64)) + EDSR(16) | 0.815±0.070 | 25.1±2.9 | 1.45±0.77 | 1.66±0.99 | 0.45±0.20 |
| NW(1, (128)) + EDSR(16) | 0.805±0.068 | 25.3±3.0 | 1.77±0.86 | 1.99±1.09 | 0.46±0.20 |
| NW(2, (64, 32)) + EDSR(15) | 0.836±0.068 | 25.3±3.0 | 1.35±0.85 | 1.56±1.08 | 0.46±0.19 |
| NW(3, (128, 64, 32)) + EDSR(14) | 0.858±0.053 | 25.7±3.0 | 1.14±0.70 | 1.36± 0.90 | 0.46±0.20 |

## 5 Conclusions

This work advances deep learning research by introducing a novel NW layer. The proposed CNN layer enables the use of sparse images as input to the CNN framework, and learn sparse image representation.

In the context of pCLE, this is the first work which proposes end-to-end deep learning-based image reconstruction with prior information to super-resolved pCLE images irregularly sampled fibre data. We proved that super-resolved pCLE images have better quality than the interpolated-based pCLE images, thus the proposed super-resolution pipeline outperforms the currently used reconstruction method.

NW layer is used as a building block in dedicated architectures (NWNets) which perform deep generalised Nadaraya–Watson kernel regression. NWNets capture data sparsity and learn reconstruction kernels for sparse data. We demonstrated that deep NWNets improve the performance of super-resolving reconstruction pipeline over an equivalent approach using standard CNNs on

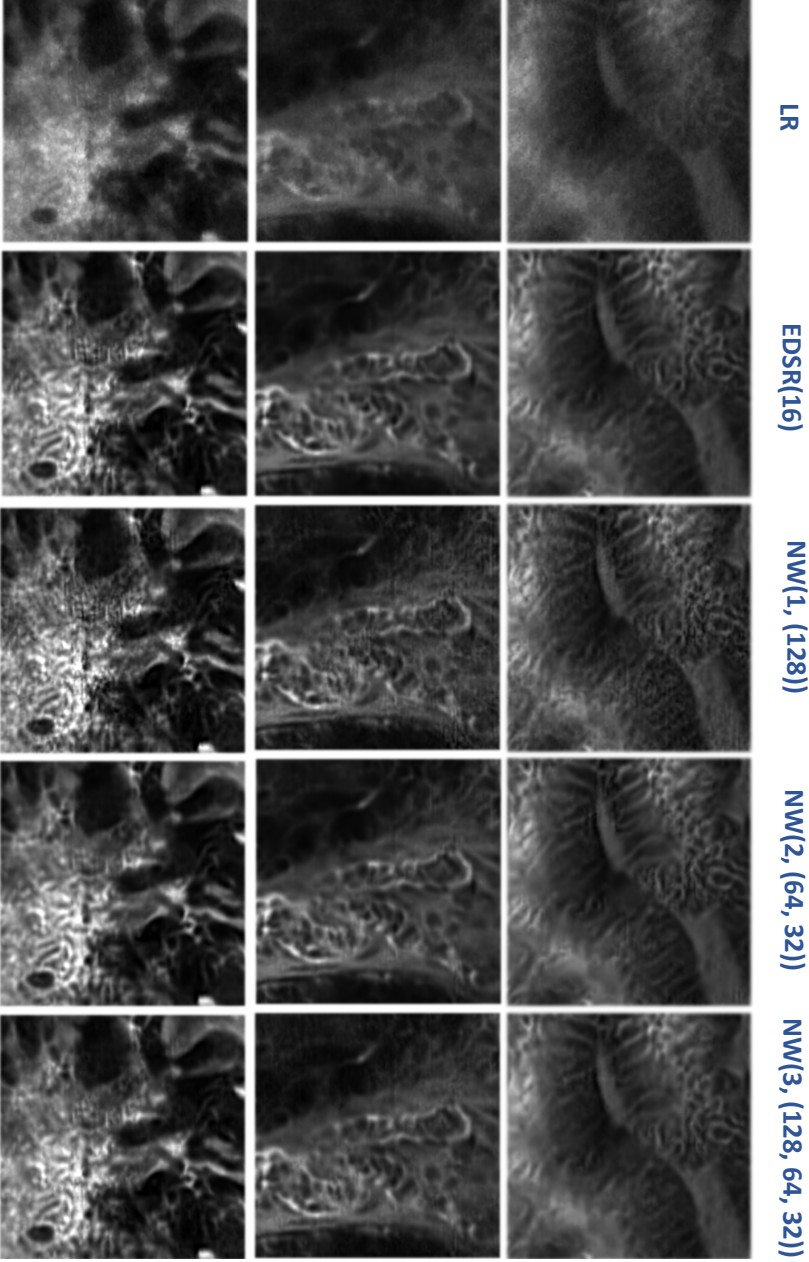

Figure 3: Sample reconstructions

interpolated input images. It leads to the conclusion that NWNets generate output features maps which capture a better representation of the input sparse data than interpolated Cartesian images.

NW layer and its extension to NWNets give an efficient way to incorporate irregularly sampled data as the input of any CNN pipeline for regularly sampled data. Deep machine learning methods for regularly sampled images, can be transferred to sparse images with a straightforward adaptation of their architecture. We have shown successful implementation of the reconstruction pipeline, which combines NWNets and EDSR to reconstruct super-resolved images form sparse input images.

Beyond pCLE, we believe that our research may benefit other applications in which data is defined on a graph structure. NWNets are computationally efficient and can readily be adapted to either graph-

in/scalar-out or graph-in/image-out. Future work will focus on developing NWNets architectures and applying NWNets to tasks such as classification.

**Acknowledgments**

This work was supported by Wellcome/EPSRC [203145Z/16/Z; NS/A000050/1; WT101957; NS/A000027/1; EP/N027078/1]. This work was undertaken at UCL and UCLH, which receive a proportion of funding from the DoH NIHR UCLH BRC funding scheme. The PhD studentship of Agnieszka Barbara Szczotka is funded by Mauna Kea Technologies, Paris, France.

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
