# OpenReview forum: "Learning from Irregularly Sampled Data with Deep Nadaraya–Watson Kernel Regression Networks (NWNet): Application to Endomicroscopy Image Reconstruction"
_MIDL.amsterdam/2018/Conference — Submitted to MIDL 2018_

### Review · AnonReviewer3 · 2018-05-09
**Neat idea but not convincing results and no appropriate comparisons**

**Rating:** 2
**Confidence:** 3

**Review:**

Authors propose an interpolation layer to address sparse inputs to CNN-based architectures. The application is image reconstruction followed by super-resolution.

Pros:
1. Problem is quite interesting and lies at the intersection of acquisition and analysis.
2. Article is well written.
3. The method is well described.
Cons:
1. Experiments are performed on synthetic data. I do not know the difficulty of getting ground truth data for this application. If it is too difficult then it is not fair to hold this against the authors. However, this aspect makes the results less reliable.
2. Numerical results do not provide a convincing picture. Authors conclude that the deepest network performs the best. However, looking at table 1, it is difficult to reach this conclusion. Keeping in mind the standard deviations, I would say they all perform just the same. The numerical differences might be due to randomness in the optimisation.
3. There are alternative ways to handle sparse input in the literature. Authors mention them but they do not compare the proposed method with them. Does the work proposed in [7] perform worse than the proposed one?
4. I do not agree with the authors on the value of the proposed network layer. A CNN that goes from the sparse input and the map directly to the super-resolution seems like the obvious benchmark to compare with. Without comparison to this simple benchmark, it is difficult to judge the value of NW-layers.
5. Article is written in an ambitious way. I would refrain from using statements such as “We propose a new paradigm…” Let the community decide that.
6. Lastly, if I am reconstructing the images for some purpose, would it not make more sense to go from sparse data directly to the task output?

**Special Issue:**

No

---

### Review · AnonReviewer2 · 2018-05-09
**The proposed approach is not fully convincing**

**Rating:** 3
**Confidence:** 3

**Review:**

Overall:
The paper considers a problem of endomicroscopy image reconstruction using a combination of Nadaraya-Watson layers and a super-resolution network. The authors propose to use a Nadaraya-Watson layer instead of a convolutional layer for sparse inputs. However, it is not convincing why the NW layer is preferable to the convnet. Especially the experiments are not conclusive (see remarks below). In general, I am not convinced by the results and claims given by the authors. I do not see a clear usefulness of the NW layers.

Strengths:
+ The paper considers an important medical imaging problem.
+ The proposed approach is a combination of newly proposed Nadaraya-Watson layer and a super-resolution model.

Remarks:
* Major
- The proposed Nadaraya-Watson layer requires a special treatment during training since it can achieve NaN or Inf, as stated by the authors. The solution given in the paper is simply assigning 0 if a numerical instability occurs. However, this sounds like a solution on the spot. It would be desirable to understand this undesirable behavior. Otherwise, any ad hoc solution could be fine and even simpler, e.g., adding a small value to the denominator (line 6 in pseudocode) or clamping values of W.
- Additionally, I do not see why the NW layer is preferable to a convolutional layer.
- The authors claim that the results in Table 1 reveals that the best performing method is NWNet(3, (128, 64, 32)) + EDSR(14). However, I do not share the authors' enthusiasm. In my opinion all methods perform similarly since any two numbers in Table 1 are always within one standard deviation (or standard error). It seems that it is hard to state that one method performs significantly better than any other method.
- I fair comparison would be to replace NWNet with standard CNN with similar capacity. However, EDSR without NWNet performs similarly to the best performing combination NWNet and EDSR, hence, it puts in question the usefulness of the proposed approach.

* Minor
- Why all names except the first author are not in bold?

**Special Issue:**

No

---

### Review · AnonReviewer1 · 2018-05-09
**This paper is interesting since authors proposed a new kind of regression layer into deep learning architecture although more scientific validations are required.**

**Rating:** 3
**Confidence:** 2

**Review:**

#Summary of paper
Authors introduced Nadaraya-Watson(NW) kernel regression to convolutional neural network as new layers for the training with sparsely cropped patches of images. They presented the experiments of image reconstruction with NW layers for probe-based confocal laser endomicroscopic (pCLE) images.

#Strength
-Introduced a new kind of layer for convolutional neural network architecture.
-Proposed the new architecture with NW layers that is based on enhanced deep residual networks for single image super-resolution [11].

#Major weakness
-Authors said that they replaced the convolutional layers with NW layers (the third paragraph from the end of section 1). However, in Fig.2 it looks just adding the NW layers before the convolutional layers.
-Reason why stacking several NW layers is ungiven. Two or three NW layers are better than one NW layer?
-The examples of pCEL images are necessary for better understanding.
-In the experimental results, PSNR and GCF for one NW layer looks better than ones for two or three NW layers.
-Is there significant difference among the comparison in Table 1?
-For the comparison of reconstruction accuracy against the number of NW layers, should the deepest of EDSR be the same?

**Special Issue:**

No

---

### Decision · Program_Chairs · 2018-05-15
**Paper93 Acceptance Decision**

Reject